# Revised Taxonomy of Rhabdoviruses Infecting Fish and Marine Mammals

**DOI:** 10.3390/ani12111363

**Published:** 2022-05-26

**Authors:** Peter J. Walker, Laurent Bigarré, Gael Kurath, Laurent Dacheux, Laurane Pallandre

**Affiliations:** 1School of Chemistry and Molecular Biosciences, The University of Queensland, St. Lucia, QLD 4067, Australia; 2Laboratory of Ploufragan-Plouzané-Niort, Technopole Brest Iroise, ANSES, 29280 Plouzané, France; laurent.bigarre@anses.fr (L.B.); laurane.pallandre@anses.fr (L.P.); 3Western Fisheries Research Center, US Geological Survey, 6505 NE 65th Street, Seattle, WA 98115, USA; gkurath@usgs.gov; 4Unit Lyssavirus Epidemiology and Neuropathology, Université Paris Cité, Institut Pasteur, 28 Rue du Docteur Roux, CEDEX 15, 75724 Paris, France; laurent.dacheux@pasteur.fr

**Keywords:** (-) RNA virus, fish rhabdovirus, taxonomy, phylogeny

## Abstract

**Simple Summary:**

The *Rhabdoviridae* is a family of viruses that includes some important pathogens of fish and marine mammals. Aspects of the taxonomic classification of fish viruses assigned to this family have recently been reviewed by the International Committee on Taxonomy of Viruses (ICTV). This paper describes the newly approved taxonomy, including the assignment of new subfamilies and new virus species. The paper also considers a taxonomic conundrum presented by viruses assigned to one group of fish rhabdoviruses (genus *Novirhabdovirus*) for which assignment to the family *Rhabdoviridae* may not be appropriate.

**Abstract:**

The *Rhabdoviridae* is a large family of negative-sense (-) RNA viruses that includes important pathogens of ray-finned fish and marine mammals. As for all viruses, the taxonomic assignment of rhabdoviruses occurs through a process implemented by the International Committee on Taxonomy of Viruses (ICTV). A recent revision of taxonomy conducted in conjunction with the ICTV *Rhabdoviridae* Study Group has resulted in the establishment of three new subfamilies (*Alpharhabdovirinae*, *Betarhabdovirinae*, and *Gammarhabdovirinae*) within the *Rhabdoviridae*, as well as three new genera (*Cetarhavirus*, *Siniperhavirus*, and *Scophrhavirus*) and seven new species for viruses infecting fish or marine mammals. All rhabdovirus species have also now been named or renamed to comply with the binomial format adopted by the ICTV in 2021, comprising the genus name followed by a species epithet. Phylogenetic analyses of L protein (RNA-dependent RNA polymerase) sequences of (-) RNA viruses indicate that members of the genus *Novirhabdovirus* (subfamily *Gammarhabdovirinae*) do not cluster within the *Rhabdoviridae*, suggesting the need for a review of their current classification.

## 1. Introduction

The *Rhabdoviridae* is a large and ecologically diverse family of viruses, members of which infect plants, vertebrates, and/or invertebrates; many rhabdoviruses are transmitted by arthropod vectors in which they replicate [1,2,3]. The negative-sense, single-stranded RNA ([-] ssRNA) genome of rhabdoviruses typically features five structural protein genes (*N*, *P*, *M*, *G*, and *L*) but commonly contains additional genes encoding nonstructural accessory proteins. For animal rhabdoviruses, the enveloped virions are typically bullet-shaped with prominent surface projections and contain a rigidly assembled tubular nucleocapsid with helical symmetry. Rhabdoviruses infecting fish and marine mammals include important pathogens that can have significant economic impacts on fish farming and environmental impacts on wild fish populations [4,5].

## 2. Virus Classification and the Virus Species Concept

The taxonomic classification of viruses is the responsibility of the International Committee on Taxonomy of Viruses (ICTV) through the authority conferred by the International Union of Microbiological Societies (IUMS) [6]. Viruses are classified into hierarchical taxonomic ranks based primarily on their evolutionary relationships with a demarcation of ranks reflecting distinguishable genetic and phenotypic characteristics. To accommodate the rapidly growing and highly diverse nature of the known virosphere, the ICTV has recently expanded the scope of virus classification to include 15 taxonomic ranks extending from realms at the highest level to the lowest level of virus species [7]. There are minimum requirements that a virus species must be assigned to a genus and that all RNA viruses employing an RNA-directed RNA polymerase (RdRP) must be assigned to the realm *Riboviria*; the assignment of viruses to intermediate taxonomic levels varies according to the availability of data on the relevant evolutionary relationships [8].

Central (but often poorly understood) aspects of virus classification are the concept of virus species and the difference between a virus and a virus species [9]. Viruses are the concrete entities that virologists study and work with daily. They are named according to common practice in the scientific literature, usually following the lead of the first report of the virus. Viruses can be isolated and purified, and their genomes can be sequenced and can infect a host in which they may cause disease. A virus species, on the other hand, is an abstract taxonomic category. One cannot isolate, purify, or sequence a virus species as they do not physically exist. Therefore, one can have an isolate or strain of a virus but not of a virus species. Virus species are named according to the rules set recently by the ICTV; the species name now must be binomial, the first word being the genus name and the second word (the species epithet) being a unique freeform identifier [10]. Unlike virus names that should never include italics (even when adopting the host species name) and are commonly abbreviated, virus species names are always italicized and should not be abbreviated. A common misconception is that the virus species name is the formal or scientific name of a virus. Unlike the practice used in other branches of biology, a virus species name is not a formal substitute for a virus name. A virus is not assigned as a species; a virus is assigned taxonomically to the rank of species.

The ICTV has also now abandoned the concept of the “type species” that was used previously to identify a virus representing a typical member of a genus [10]. The requirement for a type species was largely historical; however, it is now evident that, although viruses selected to represent the type species may have been either the first discovered or most studied member of a genus, they were quite often not necessarily typical of all members of the genus. With the increasing importance of phylogenetic relationships in defining genera, it is the relationship between the sequences of the member viruses, and not the sequence of any particular member, that defines a genus. Consequently, rather than a “type species”, the ICTV now requires the identification of at least one “exemplar” virus sample for each virus species and a GenBank nucleotide sequence deposition for each exemplar virus. Typically, a complete or near-complete coding sequence is required for classification of a virus.

## 3. The Taxonomic Structure of the Family *Rhabdoviridae*

Rhabdoviruses were named originally for the apparently unique and characteristic morphology of the virions (*rhabdos* [Greek] = rod). Animal rhabdoviruses were described as bullet- or cone-shaped, and plant rhabdoviruses as rod-shaped with two rounded ends. The family *Rhabdoviridae* was established by the ICTV in 1976 comprising only two formally approved genera (*Vesiculovirus* and *Lyssavirus*) and 15 viruses assigned to species [11]. Following ratification by the ICTV membership in February 2022, the family *Rhabdoviridae* currently comprises some 45 genera and 265 species approved for viruses infecting or detected in plants, invertebrates (arthropods, nematodes), or vertebrates (mammals, amphibians, reptiles, birds, fish) [12]. Within the *Rhabdoviridae*, there are now three subfamilies (*Alpharhabdovirinae*, *Betarhabdovirinae*, *Gammarhabdovirinae*), two of which contain viruses of fish and marine mammals as detailed below. The family is also now assigned to higher taxonomic ranks: realm *Riboviria*, kingdom *Orthornavirae*, phylum *Negarnaviricota*, subphylum *Haploviricotina*, class *Monojiviricetes*, order *Mononegavirales* [13].

The taxonomic structure within the *Rhabdoviridae* is based primarily on evolutionary relationships determined by phylogenetic analysis of rhabdovirus L protein (RdRP) amino acid sequences [12]. The L protein is the most highly conserved of the rhabdovirus proteins, allowing relationships to be determined across the extent of this diverse family, and indeed beyond to include other [-] ssRNA viruses. The use of a single genetic marker for the identification of evolutionary relationships is justified by the absence or extremely rare occurrence of genetic recombination in rhabdoviruses [1].

The demarcation of taxa within the family generally reflects the ecological context of rhabdovirus evolution through which clusters of closely related viruses usually share similar categories of host and/or arthropod vector. The inherently dynamic nature of rhabdovirus genome evolution is also a consideration in the demarcation of genera [1]. Although core structural protein genes (particularly *N*, *P*, *M*, and *L*) are invariably retained, accessory genes may be gained de novo and subsequently lost during rhabdovirus genome evolution and, as a consequence, viruses clustering at the genus level usually share similar genome architectures [1,14]. Ultimately, viruses assigned to each subfamily or genus must be monophyletic based on L protein sequences but the points of demarcation can be somewhat arbitrary, depending on whether the chosen approach is based on “lumping” or ”splitting”.

The demarcation of rhabdovirus species requires consideration of several criteria typically based on amino acid sequence identities of structural proteins, natural host/vector associations, and, when available, the virus neutralization phenotype. Pathogenicity is considered to be too much dependent on environmental and host factors, and too easily modified by mutation, to be a useful consideration in species demarcation. Nevertheless, despite the variable nature of this approach as applied to viruses assigned to different genera, it can be argued that clusters of viruses representing the species taxon do occur naturally and can be identified by clear discontinuities in the amino acid or nucleotide sequence identity profiles of virus isolates. Although potentially influenced by the variation in the rates of evolution across the family, isolates of viruses assigned to individual rhabdovirus species typically display the protein amino acid sequence divergence of <10% in the L and N proteins, and <15% in the G protein.

## 4. The Subfamily Alpharhabdovirinae

The *Alpharhabdoviridae* is the largest subfamily currently assigned within the *Rhabdoviridae*, comprising 31 genera and 189 approved species for viruses infecting mammals, amphibians, reptiles, birds, fish, insects, ticks, or nematodes [12]. Viruses assigned to the *Alpharhabdovirinae* have sometimes been referred to informally as dimarhabdoviruses (dipteran-mammalian rhabdoviruses), but this term does not correctly capture the extent of ecological diversity within the clade [15]. The subfamily now includes five genera for viruses infecting teleost fish and marine mammals (Table 1). Each genus forms a monophyletic clade of viruses based on the alignment of L protein sequences (Figure 1). Demarcation of the genera is based on considerations of the different ecological contexts in which the viruses have evolved and the genetic distances between the clades. The assignment of genera may potentially change in the future as more viruses are discovered and evolutionary relationships are more clearly delineated.

The genus *Sprivivirus* currently includes two species: *Sprivivirus cyprinus* and *Sprivivirus esox*. The species *Sprivivirus cyprinus* is assigned for spring viraemia of carp virus (SVCV). SVCV causes a lethal hemorrhagic disease in cyprinids, particularly common carp (*Cyprinus carpio*). The exemplar sample (VR-1390; GenBank U18101) was isolated from diseased common carp in former Yugoslavia more than 50 years ago [16]. Various genetic lineages of SVCV have since been reported from Europe, Asia, and the Americas [17]. The species *Sprivivirus esox* is assigned for pike fry rhabdovirus (PFRV). The exemplar sample (F4; Genbank FJ872827) was isolated in 1972 during an outbreak of hemorrhagic disease (red disease) in cultured northern pike (*Esox lucius*) in the Netherlands [18]. Two other viruses are classified as members of the species *Sprivivirus esox*. Grass carp rhabdovirus (GCRV V76; GenBank KC113518) and tench rhabdovirus (TRV S64; GenBank KC113517) were each isolated in Germany in 1982 from cyprinids of two species (*Ctenopharyngodon idella* and *Tinca tinca*, respectively).The amino acid sequence divergence (p-distance) amongst 48 SVCV isolates sampled from fish of different species in Europe, Asia, and North America over a period of more than 50 years is ≤2.7% in the L protein, ≤3.4% in the N protein, and ≤8.7% in the G protein (Table 1). In contrast, the amino acid sequence divergence between all SVCV isolates and the three isolates assigned to the species *Sprivivirus esox* (PFRV, GCRV, and TRV) is ≥11.8% in L, ≥8.2% in N, and ≥25.4% in G (Table 1). The amino acid sequence divergence amongst the three viruses assigned to the species *Sprivivirus esox* is ≤6.7% in L, ≤5.6% in N, and ≤16.7% in G (Table 1). Although this range is greater than observed for SVCV isolates, it falls below the sequence divergence threshold separating viruses representing the two different species, justifying the assignment of all three isolates to the same sprivivirus species. Although virus neutralization and host range/susceptibility data are not available, the limited sequence divergence suggests that PFRV, GCRV, and TRV may be considered to be isolates of the same virus. Divergence data for all available sprivivirus L, N, and G amino acid sequences are shown in Appendix A.

The genus *Perhabdovirus* currently includes four species: *Perhabdovirus perca*, *Perhabdovirus trutta*, *Perhabdovirus anguilla*, and *Perhabdovirus leman*. The species *Perhabdovirus perca* is assigned perch rhabdovirus (PRV). The exemplar sample (Dorson; GenBank JX679246) was isolated in 1981 from European perch (*Perca fluviatilis*) in France [19]. PRV has also been isolated from pikeperch (*Sander lucioperca*) in Belgium [20]. The species *Perhabdovirus trutta* is assigned for lake trout rhabdovirus (LTRV). The exemplar sample (903/87; GenBank AF434991) was isolated in 1987 from moribund brown trout (*Salmo trutta*) fingerlings from Finland [21]. Several other isolates of this virus, but sometimes named sea trout rhabdovirus, have been reported from France and Switzerland. The species *Perhabdovirus anguilla* is assigned for eel virus European X (EVEX), which was first isolated in 1976 in Japan from a shipment of European eels (*Anguilla anguilla*) from France [22,23]. The exemplar sample of EVEX (CV1153311; GenBank FN557213) was isolated from farmed European eel in the Netherlands in 1992. In 1974, a rhabdovirus was isolated from young American eel (*Anguilla rostrata*) imported from Cuba to Japan and named eel virus American (EVA) [23,24]. EVA and EVEX are morphologically, serologically, and genetically highly similar and are considered to be strains of the same virus [23,25]. Multiple isolates of EVEX from European eel in Europe and Japan have been sequenced. The new species *Perhabdovirus leman* is assigned for Leman virus (LEMV). The exemplar sample (LEMV 18/193; GenBank MN963996) was isolated in 1999 from symptomatic wild young perch collected from Lake Leman in France [26]. The amino acid sequence divergence amongst viruses assigned to the same perhabdovirus species is ≤5.2% in L, ≤6.1% in N, and ≤11.6% in G. The amino acid sequence divergence between viruses assigned to different perhabdovirus species is ≥13.1% in L, ≥15.9% in N, and ≥16.7% in G (Table 1; Appendix A).

The new genus *Cetarhavirus* includes two species for viruses infecting aquatic mammals (Cetacea): *Cetarhavirus lagenorhynchus* and *Cetarhavirus phocoena*. The new species *Cetarhavirus lagenorhynchus* is assigned for dolphin rhabdovirus (DRV). The exemplar sample (pxV1; GenBank KF958252) was isolated from a white-beaked dolphin (*Lagenorhynchus albirostris*) stranded on the Dutch island of Schiermonnikoog in 1992 [27]. A neutralizing antibody to DRV has been detected in various cetaceans (dolphins, porpoises, whales) and pinnipeds (seals) sampled from the coast of northwest Europe or the Mediterranean Sea [27,28]. The new species *Cetarhavirus phocoena* is assigned for harbor porpoise rhabdovirus (HPRV). The exemplar sample (WVL17017A; GenBank MN103537) was isolated from a harbor porpoise (*Phocoena phocoena*) stranded off the coast of Alaska in 2013 [29]. These are the only reported samples of the viruses at this time. The amino acid sequence divergence between the exemplar samples of DRV and HPRV is 31.3% in L, 31.8% in N, and 56.1% in G (Table 1; Appendix A).

The new genus *Siniperhavirus* currently includes two species: *Siniperhavirus zoarces* and *Siniperhavirus chuatsi*. The new species *Siniperhavirus zoarces* is assigned for eelpout rhabdovirus (EPRV). The exemplar sample (FSK0523; GenBank KR612230) was detected by high-throughput sequencing in samples of eelpout (*Zoarces viviparous*) collected during mass fish mortalities near Stockholm, Sweden, in 2014 [30]. This is the only virus currently assigned to this species. The new species *Siniperhavirus chuatsi* is assigned for Siniperca chuatsi rhabdovirus (SCRV). The exemplar sample (GenBank DQ399789) was isolated in 1997 from mandarin fish (*Siniperca chuatsi*) collected in Guangdong Province, China [31,32]. Several other viruses have been assigned as members of the species *Siniperhavirus chuatsi*. Hybrid snakehead rhabdovirus (C1207; GenBank KC519324) was isolated in 2012 from a moribund hybrid snakehead fish (*Channa maculata* × *Channa argus* cross) collected in Guangdong Province, China [33]. HSRV (Xingtan; GenBank KP876483) was subsequently isolated from hybrid snakehead fish from the same province of China in 2014 and shown to infect mandarin fish [34]. A third HSRV isolate (SHVV-02019; GenBank MW291462) was obtained from snakehead fish (*Channa argus*) in China in 2019. Chinese rice-field eel rhabdovirus (CrERV) (GenBank MH319839) was isolated from diseased Asian swamp eels (*Monopterus albus*) collected from Hubei Province, China, in 2017 [35]. Micropterus salmoides rhabdovirus (MSRV) was first discovered in juvenile mandarin fish collected in Guangdong Province, China, in 2011 [36]. A second sample (YH01; GenBank MK397811) was isolated from moribund largemouth bass (*Micropterus salmoides*) collected in Zhejiang Province, China, in 2017 [37]. A third isolate of the virus (FJ985; GenBank MT818233) was obtained from China in 2019. The amino acid sequence divergence amongst all of these isolates (SCRV, HSHV, CrERV, MSRV) is low (≤3.6% in L, ≤6.8% in N, and ≤10.6% in G) and they should all be considered to be variants of the same virus. The amino acid sequence divergence between viruses assigned to the two different siniperhavirus species is ≥28.8% in L, ≥33.7% in N, and ≥50.0% in G (Table 1; Appendix A).

The new genus *Scophrhavirus* currently includes two species: *Scophrhavirus maximus* and *Scophrhavirus chanodichthys*. The new species *Scophrhavirus maximus* is assigned for Scophthalmus maximus rhabdovirus (SMRV). The exemplar sample (GenBank HQ003891) was isolated from turbot fish (*Scophthalmus maximus*) with signs of hemorrhagic disease collected from Shandong Province, China [38]. The new species *Scophrhavirus chanodichthys* is assigned for Wuhan redfin culter dimarhabdovirus (WhRCDRV). The exemplar sample (DSYS6218; GenBank MG600013) was detected by high-throughput sequencing in redfin culter (*Chanodichthys erythropterus*) collected in Hubei Province, China [39]. These are the only reported samples of the viruses at this time. The amino acid sequence divergence between the exemplar samples of SMRV and WhRCDRV is 46.5% in L, 62.6% in N, and 71.0% in G (Table 1; Appendix A).

## 5. Subfamily Gammarhabdovirinae

The *Gammarhabdovirinae* is the smallest subfamily in the *Rhabdoviridae*, currently comprising only a single genus for viruses infecting teleost fish. The genus *Novirhabdovirus* currently includes four species: *Novirhabdovirus salmonid*, *Novirhabdovirus piscine*, *Novirhabdovirus hirame*, and *Novirhabdovirus snakehead*.

The species *Novirhabdovirus salmonid* is assigned for infectious hematopoietic necrosis virus (IHNV). The exemplar sample (WRAC; GenBank L40883) was isolated in 1982 from diseased rainbow trout (*Oncorhynchus mykiss*) in Idaho, USA [40]. IHNV causes economically important disease in a wide variety of salmonid fish species. The virus is enzootic in coastal areas and river systems throughout western North America but has spread to Asia and Europe through translocation of infected stock [41]. IHNV is resolved globally into five major genogroups (U, M, L, J, and E) and several subgroups that vary in geography and host specificity [42]. The amino acid sequence divergence across all available IHNV isolates of all genotypes is ≤2.4% in L, ≤12.6% in N, and ≤10.2% in G (Table 1).

The species *Novirhabdovirus piscine* is assigned for viral hemorrhagic septicaemia virus (VHSV). The exemplar sample (Fil3; GenBank Y18263) was isolated in 1962 from diseased rainbow trout (*Oncorhynchus mykiss*) in Denmark [43]. VHSV is considered to be a pathogen of major economic and environmental importance. It has been isolated from or detected in a wide range of teleost fish in Europe, Asia, and North America [44]. VHSV falls into four major genotypes (I–IV) and various sub-types with naturally confined geographic distributions [45,46,47]. The amino acid sequence divergence across all available VHSV isolates of all genotypes is ≤7.1% in L, ≤11.1% in N, and ≤11.5% in G (Table 1).

The species *Novirhabdovirus hirame* is assigned for hirame rhabdovirus (HIRRV). First detected in Japan in 1984, HIRRV causes a hemorrhagic disease characterized by congestion of the gonads and the accumulation of ascitic fluid [48]. The exemplar sample (CA9703; GenBank AF104985) was isolated in 1997 from Japanese flounder (*Paralichthys olivaceus*) cultured in the Republic of Korea [49,50]. HIRRV occurs in a wide range of marine fish in several countries in East Asia [49,51,52]. HIRRV has also caused mortalities in freshwater fish in Europe, possibly as a result of translocation from East Asia [53]. The amino acid sequence divergence across the small number of available HIRRV isolates is 0.4% in L, 0.8% in N, and ≤1.2% in G (Table 1).

The species *Novirhabdovirus snakehead* is assigned for snakehead rhabdovirus (SHRV). The exemplar sample (GenBank AF147498) was isolated in 1986 from snakehead fish (*Ophicephalus struatus*) in Thailand [54,55] with a disease characterized by necrotic ulcerations. The disease was reported in wild and cultured snakehead fish in several countries of Southeast Asia and various other organisms (viruses, bacteria, fungi, and parasites) have been found in association with diseased fish [55,56]. Sequences are currently available only for the exemplar sample of SHRV.

The amino acid sequence divergence between viruses assigned to different novirhabdovirus species is ≥15.1% in L, ≥35.8% in N, and ≥22.9% in G (Table 1). Divergence data for all available novirhabdovirus L, N, and G amino acid sequences are shown in Appendix A.

Classification of the genus *Novirhabdovirus* presents somewhat of a taxonomic conundrum. Historically, the novirhabdoviruses have been placed taxonomically within the *Rhabdoviridae* as they share many of the characteristics of classical rhabdoviruses. Virions are enveloped, bullet-shaped particles with clear surface projections and a helical nucleocapsid [57,58,59]. The novirhabdovirus genome comprises homologs of the five rhabdovirus structural protein genes (*N*, *P*, *M*, *G*, and *L*) as well as an additional gene (*NV*) encoding a unique nonstructural protein that has been shown to be involved in pathogenesis and evasion of host immune responses [60,61,62]. Conserved transcription initiation and transcription termination sequences flank each gene and, as occurs typically in all alpharhabdoviruses, the transcription termination sequences feature a conserved run of seven uridine residues [40,63]. Importantly, novirhabdoviruses encode a single type I transmembrane glycoprotein (G) that is structurally homologous with the G proteins of other animal rhabdoviruses, featuring a unique set of cysteine residues that stabilize the pre- and post-fusion-folded structures of the protein [64,65,66,67]. Nevertheless, phylogenetically, the novirhabdoviruses sit separately from all other members of the *Rhabdoviridae*.

Evolutionary analysis using the complete L protein (RdRP) sequences of viruses representing families in the order *Mononegavirales* indicates that novirhabdoviruses do not cluster monophyletically with other rhabdoviruses but tend to cluster with members of the families *Paramyxoviridae*, *Pneumoviridae*, and *Filoviridae* (Figure 2; Appendix A) [68]. The RdRP is considered to be the most useful marker for mapping viral evolutionary history as it the most highly conserved sequence element and is indicative of the core replicating lineage. Indeed, there is evidence that all viral RdRPs and reverse transcriptases are monophyletic [69,70], leading to the creation by the ICTV of the realm *Riboviria* to accommodate all viruses with RNA genomes, the phylum *Negarnaviricota* for all [-] ssRNA viruses, and various other intermediate and subsidiary taxonomic ranks [7]. Evolutionary analyses based on the phylogeny of mononegavirus L proteins cannot, therefore, be disregarded.

A possible explanation for the apparent dichotomy between the novirhabdovirus virion structure and RdRP-based phylogeny is that paramyxovirus, pneumovirus, and filovirus glycoprotein genes were introduced into an ancestral [-] ssRNA virus by recombination or lateral gene transfer after the novirhabdovirus lineage had diverged from that of other rhabdoviruses. Indeed, this appears to be supported by alignments of the type I transmembrane glycoprotein sequences of members of the family *Chuviridae* with the vesicular stomatitis Indiana virus (VSIV) and novirhabdovirus G proteins (Figure 3), which all share subsets of the same conserved set of cysteine residues. Although genetic recombination occurs very rarely in mononegaviruses, several other possible recombination events have been reported previously [72,73,74,75,76]. Indeed, the structural homology between the VSIV G protein and herpes simplex virus 1 glycoprotein gB indicates that they also have a common evolutionary origin involving an ancient recombination event [77]. Whatever the exact mechanism, it appears that some form of either recombination or lateral gene transfer would be required to arrive at the phylogenetic relationships determined using RdRP-based sequence alignments.

Nevertheless, we suggest that removal of the novirhabdoviruses from the *Rhabdoviridae* requires careful consideration. For example, do we know that alignments used to generate phylogenetic trees from such distantly related RdRP sequences are sufficiently reliable to be confident of the deep nodes? Certainly, alignments do vary according to the algorithms employed and parameters selected, and the reliability of sequence alignments used to infer deeply rooted phylogenies linking all RNA viruses has been questioned [69,78]. Moreover, it has been recognized that “even a correct and informative alignment does not guarantee correct phylogenetic reconstruction due to the technical limitations of the software, systematic biases of the available evolutionary models, and the fundamentally random nature of sequence divergence” [79] and that evolutionary relationships inferred from phylogenetic analysis need to take account of associated biological data [79]. The structural homology of hallmark genes such as capsid proteins has been used to define taxonomic relationships for higher taxonomic ranks of some DNA viruses for which there are no common genes displaying evident sequence homology [80,81]. In the absence of confidently reliable sequence alignments, should virion structural homology also be a consideration in the demarcation of some lower taxa? A global view of the molecular and structural properties of novirhabdoviruses may provide a more informative and useful guide to their taxonomic classification than an analysis based only on RdRP sequence alignments.

## 6. Conclusions

As the result of proposals approved by the ICTV in February 2022, all currently known rhabdoviruses infecting fish or marine mammals for which complete or near-complete genome sequences are available have now been assigned taxonomically to the ranks of genus and species. This is a significant advance as many of these viruses and their complete coding sequences have been known for years but they were not accommodated in the previous taxonomic structures. All rhabdovirus species, including those infecting fish or marine mammals, have also been named or renamed to comply with the binomial format approved by the ICTV in 2021. Importantly, these taxonomic assignments and the renaming of species have no bearing on the names of the viruses themselves, which will continue to follow the long-standing practice of reflecting the common usage adopted in the scientific literature.

In a practical sense, changes to virus taxonomy can be very disruptive to governments and industry as the taxonomic classification is often embedded in regulations and legislation that are intended to ensure timely reporting of detected pathogens and limit the spread of viral diseases. Nevertheless, virus taxonomy is by nature a complex tapestry that develops and evolves as knowledge of the virosphere expands. To remain useful and relevant, taxonomic classifications must adapt as new viruses are discovered and the evolutionary and ecological relationships between viruses are better understood. This is particularly so in the age of metagenomic sampling and high-throughput sequencing that has seen very rapid growth in the number of viral genome sequences and concomitant expansion in the number of newly assigned taxa. It is hoped that this review will assist by promulgating the current taxonomic assignments and by explaining some underlying principles of virus taxonomy that are often misunderstood by members of the scientific community.

## Figures and Tables

**Figure 1 animals-12-01363-f001:**
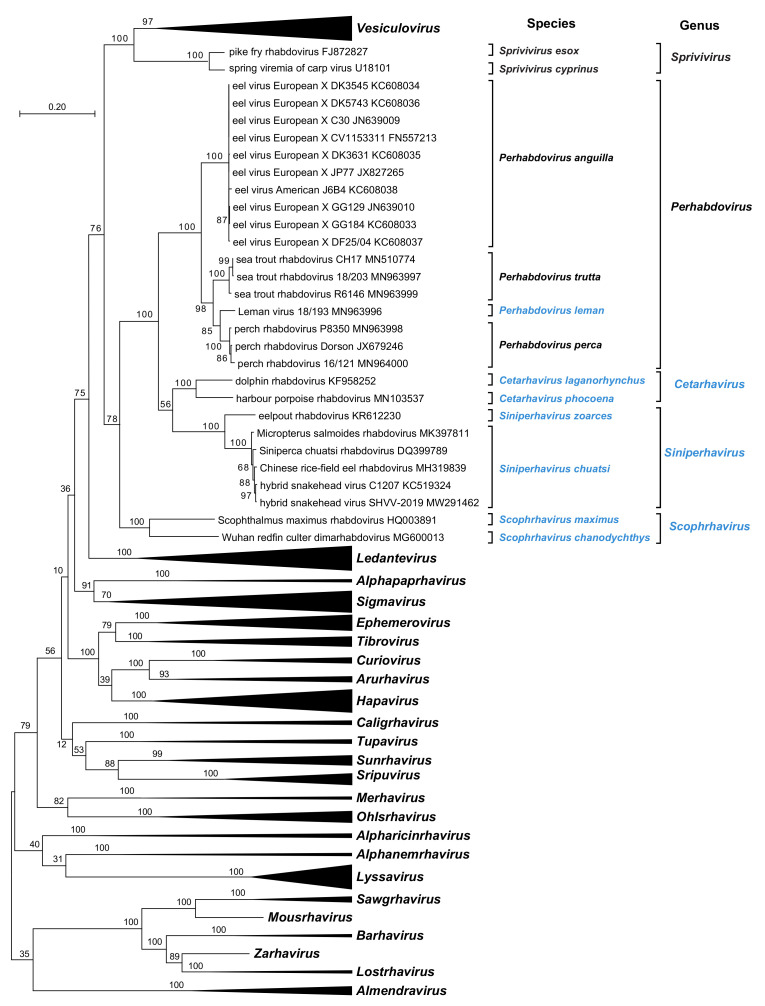
The evolutionary history was inferred from a Clustal Omega alignment of 191 complete L protein sequences of animal rhabdoviruses currently assigned to species in the subfamily *Alpharhabdovirinae*. Phylogenetically informative sites were selected from the alignment using Gblocks resulting in 1029 positions in the final dataset. The tree was inferred in MEGA7 by using the maximum likelihood method based on the WAG + Γ amino acid substitution model. The tree with the highest log likelihood (−151,963.70) is shown. The percentage of trees in which the associated taxa clustered together is shown next to the branches. Initial tree(s) for the heuristic search were obtained automatically by applying Neighbor-Joining and BioNJ algorithms to a matrix of pairwise distances estimated using a JTT model, and then selecting the topology with the superior log likelihood value. The tree is drawn to scale, with branch lengths measured in the number of substitutions per site. Bootstrap values (100 iterations) are shown for each node. Newly assigned genera and species are shown in blue font.

**Figure 2 animals-12-01363-f002:**
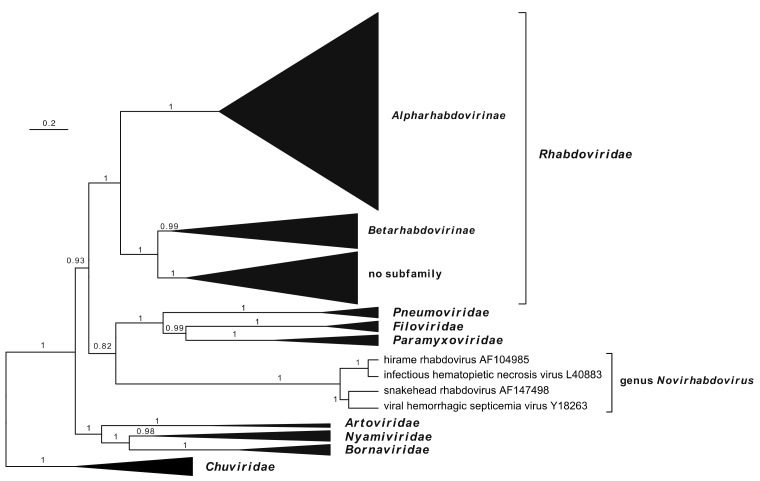
Maximum clade credibility (MCC) tree inferred from MAFFT alignments of full-length rhabdovirus L sequences of viruses representing all four members of the genus *Novirhabdovirus*, as well as all other genera in the *Rhabdoviridae* (85 sequences), and members of the families *Paramyxoviridae* (4 sequences), *Pneumoviridae* (4 sequences), *Filoviridae* (4 sequences), *Bornaviridae* (4 sequences), *Nyamiviridae* (3 sequences), *Atroviridae* (2 sequences), and *Chuviridae* (6 sequences). Ambiguously aligned regions were removed from the alignment using TrimAl [71], resulting in a final alignment length of 916 amino acids. The MCC tree was inferred in BEAST.v1.10.4 by using the Whelan and Goldman (WAG) model of amino acid substitutions, the gamma + invariant sites model of site heterogeneity, and a strict molecular clock (coalescent: constant size) with a random starting tree to perform 10 million MCMC runs. The analysis was sampled at every 10000 states. Tree Annotator v1.10.4 was used to output the results of the MCC tree model and calculate posterior probabilities with a burn-in of 1 million states. FigTree was then used to plot the MCC phylogenetic tree. The tree is drawn to scale, with branch lengths measured in the number of substitutions per site and rooted on the chuvirus clade. Posterior probability values are shown for each branch. Maximum-likelihood trees inferred from the same amino acid sequence alignment are shown in Appendix A. Family, subfamily and genus level taxonomic assignments are shown in bold.

**Figure 3 animals-12-01363-f003:**
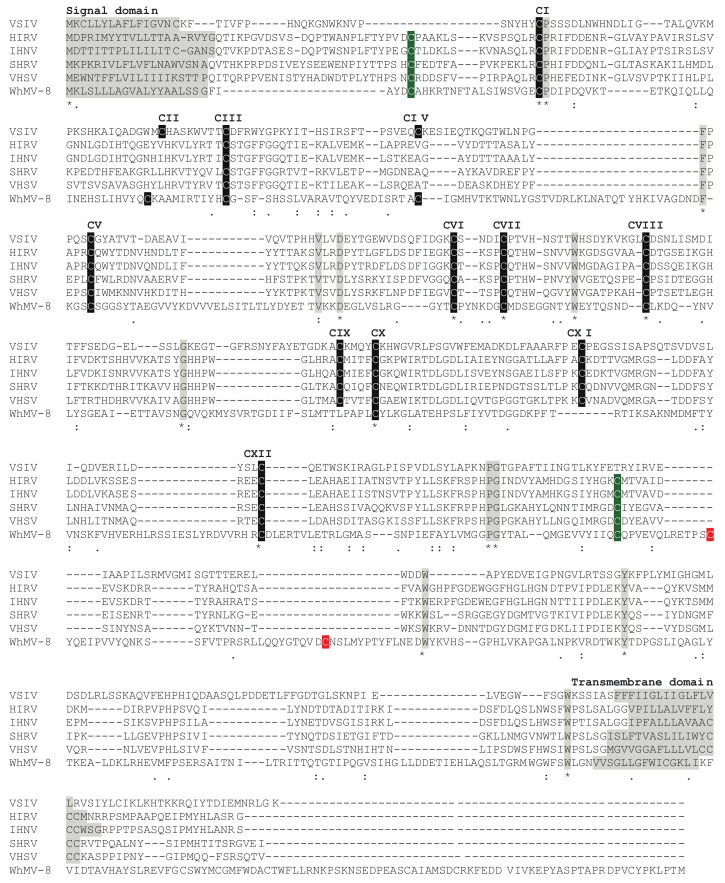
Clustal Omega alignment of the G proteins of vesicular stomatitis Indiana virus (VSIV), hirame rhabdovirus (HIRV), infectious hematopoietic necrosis virus (IHNV), snakehead rhabdovirus (SHRV), viral hemorrhagic septicemia virus (VHSV), and the chuvirus Wuhan mosquito virus 8 (WhMV-8). The signal domain and transmembrane domains of these class I transmembrane glycoproteins are indicated. Twelve cysteine residues in VSIV (CI-CXII) form six disulfide bridges (CI-CXII, CII-CIV, CIII-CV, CVI-CVII, CVIII-CX, and CIX-CXI), one of which (CII-CIV) is absent in the novirhabdovirus G proteins [64,67,82]. The WhMV-8 G protein lacks the CIX-CXI disulfide bridge, shares two additional cysteine residues with the novirhabdoviruses (shaded green), and has two unique cysteine residues (shaded red) in the ectodomain that may also form a disulfide bridge.

**Table 1 animals-12-01363-t001:** Taxonomic classification of rhabdoviruses infecting fish or marine mammals (2022).

Subfamily	Genus	Species	Exemplar Virus	Abbrev.	Sample ^§^	GenBank Accession	Intra-Species Divergence (%) ^#^			Inter-Species Divergence (%) ^#^		
							L	N	G	L	N	G
*Alpharhabdovirinae*	*Sprivivirus*	*Sprivivirus cyprinus*	spring viremia of carp virus	SVCV	VR-1390	U18101	≤2.7	≤3.4	≤8.7	≥11.8	≥8.2	≥25.4
		*Sprivivirus esox*	pike fry rhabdovirus	PFRV	F4	FJ872827	≤6.7	≤5.6	≤16.7			
	*Perhabdovirus*	*Perhabdovirus perca*	perch rhabdovirus	PRV	Dorson	JX679246	≤4.5	≤2.7	≤3.3	≥13.1	≥15.9	≥16.7
		*Perhabdovirus trutta*	lake trout rhabdovirus	LTRV	903/87	AF434991	≤5.2	≤6.1	≤11.6			
		*Perhabdovirus anguilla*	eel virus European X	EVEX	CV1153311	FN557213	≤2.2	≤1.7	≤4.1			
		*Perhabdovirus leman*	Leman virus	LEMV	18/193	MN963996	n.a.	n.a.	n.a.			
	*Cetarhavirus*	*Cetarhavirus lagenorhynchus*	dolphin rhabdovirus	DRV	pxV1	KF958252	n.a.	n.a.	n.a.	31.3	31.8	56.1
		*Cetarhavirus phocoena*	harbour porpoise rhabdovirus	HPRV	WVL17017A	MN103537	n.a.	n.a.	n.a.			
	*Siniperhavirus*	*Siniperhavirus zoarces*	eelpout rhabdovirus	EPRV	FSK0523	KR612230	n.a.	n.a.	n.a.	≥28.8	≥33.7	≥50.0
		*Siniperhavirus chuatsi*	Siniperca chuatsi rhabdovirus	SCRV	n.a.	DQ399789	≤3.6	≤6.8	≤10.6			
	*Scophrhavirus*	*Scophrhavirus maximus*	Scophthalmus maximus rhabdovirus	SMRV	n.a.	HQ003891	n.a.	n.a.	n.a.	46.5	62.6	71.0
		*Scophrhavirus chanodichthys*	Wuhan redfin culter dimarhabdovirus	WhRCDRV	DSYS6218	MG600013	n.a.	n.a.	n.a.			
*Gammarhabdovirinae*	*Novirhabdovirus*	*Novirhabdovirus salmonid*	infectious haematopoietic necrosis virus	IHNV	WRAC	L40883	≤2.4	≤12.6	≤10.2	≥15.1	≥35.8	≥22.9
		*Novirhabdovirus piscine*	viral haemorrhagic septicaemia virus	VHSV	Fil3	Y18263	≤7.1	≤11.1	≤11.5			
		*Novirhabdovirus hirame*	hirame rhabdovirus	HIRRV	CA9703	AF104985	0.4	0.8	≤1.2			
		*Novirhabdovirus snakehead*	snakehead rhabdovirus	SHRV	n.a.	AF147498	n.a.	n.a.	n.a.			

^§^ Samples of exemplar viruses listed by ICTV (https://talk.ictvonline.org/taxonomy/vmr/m/vmr-file-repository/13181 (accessed on 18 October 2021)). ^#^ p-distances estimated in MEGA 6.0 using all available full-length amino acid sequences. n.a. = not applicable (only one complete sequence currently available).

## Data Availability

Not applicable.

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
