# Peer review of "Revised Taxonomy of Rhabdoviruses Infecting Fish and Marine Mammals"

_animals, 2022, doi:10.3390/ani12111363_

Round 1

Reviewer 1 Report

Title: Revised taxonomy of rhabdoviruses infecting fish and marine mammals

Corresponding Author: Peter J Walker

General Comment: 

This manuscript describes the phylogenetic ordination of viruses of fish and marine mammals within the Rhadboviridae family.  The manuscript is well written and provides a review of newly ICTV approved taxonomy for this virus family with a specific focus on those members infecting fish and marine mammals. The manuscript clearly describes key features of the revised taxonomy which include the formation three new genera, the placement of multiple new species, and highlights the potential misplacement of the Novirhabdovirus genus among rhabdoviruses.  Publication of the manuscript is recommended with the following comments provided for the authors consideration.

Specific comments:

While it is greatly appreciated that the authors provided thoughts and speculations on those mechanisms required for the Novirhabdovirus genus to arrive at its current phylogenetic placement, it would be welcomed if the authors could explain how the revised placement of the Novirhabdovirus genus compares with previous taxonomies.  Is this a significant departure than what has been modeled previously?  I suspect that the novirhabdoviruses have always been phylogenetically far basal to the other genera yet as currently written one could presume that the revised phylogeny is a considerable deviation from previous phylogenies.  A paragraph providing the historical context to the placement of the Novirhabdovirus genus would be beneficial to the understanding of how the most recent taxonomy compares with previous taxonomic assessments.

The authors indicate that the Novirhabdovirus genus is a “taxonomical conundrum” because while L protein sequences suggest a non Rhabdoviridae origin, the novirhabdoviruses share many similarities with Rhabdoviruses.  In fact the authors reveal conservation in the glycoprotein gene, as evident in an alignment (Figure 3), yet it is curious why the authors did not include a phylogeny based on the G-protein sequences as a more direct comparison with the Novirhabdovirus genus placement obtained using RdRp sequences.

Line 376: delete “the” prior to “L protein”

Author Response

We thank the reviewer for their comments on the manuscript.

  1. Previous (and continuing current) taxonomic placement of viruses in the genus Novirhabdovirus within the Rhabdoviridae is based on the range of similar characteristics that are outlined in Section 5 of the paper i.e., principally the common morphology, homologs of the major structural protein genes, conserved transcription termination sequences and glycoprotein structure. We have now modified the introduction to the paragraph describing the properties to make this clearer.
  2. The reviewer has suggested the use of G protein phylogeny to demonstrate the placement of the novirhabdoviruses within the Rhabdoviridae. However, we point out that, although the G proteins share some structural characteristics (e.g., conserved cysteine residues, transmembrane domains and signal domains), there are no conserved sequence motifs and the level of overall G protein sequence identity across the Rhabdoviridae is extremely low. Meaningful phylogenetic reconstruction using G protein sequences is therefore not possible.

Reviewer 2 Report

Concerning my previous referee´s statement, the information provided by editor clearly makes my objections non-valid. It is a review, and it is submitted to a Special Issue which has a title that makes this a relevant review contribution. I am certainly not a virus taxonomist, and cannot judge regarding taxonomical issues. The article gives good rationale for revision of taxonomy and the need for virus taxonomy to adapt, as new viruses are isolated and methods are improved. However, the description of the fish pathogens (which are abundant in this taxonomic group) IHNV, VHSV, HIRRV, etc., is adequate and correct. In conclusion, I no longer see any reason to object the publication of this paper.

Author Response

We thank the reviewer for their comments.

The reviewer does not request any modifications to the manuscript.

Reviewer 3 Report

The article "Revised taxonomy of rhabdoviruses infecting fish and marine mammals" describe the Rhabdoviridae family, new taxonomy and new genuses - very importatnt topic in aquatic world. All necessary information are include in the text and well described. Figures and tables are helpful and also well described. The list of references contain a lot of positions, which show how authors well prepared this article.

In my opinion this article can be accept in present form.

specific comments:

  1. What is the main question addressed by the research? Te main question is what is new in Rabdoviridae family taxonomy, how it influence on current classification and article describe principles of virus taxonomy.  
  2. Do you consider the topic original or relevant in the field, and if
    so, why? I consider that topic is original and relevant in the field, because describes new changes in Rhabdoviridae taxonomy and describes Rhabdoviridae family well. This is important for resarchers who work with fish viruses, especially viruses from Rhabdoviridae family. Authors gathered many information about principles of virus taxonomy and information about Rabdoviridae family in one place.  
  3.  What does it add to the subject area compared with other published
    material? Authors describe changes in virus taxonomy, which appaerd in 2021. No everyone know about these changes, so the topic is relevant. 
  4. What specific improvements could the authors consider regarding the
    methodology? It is hard to say about methodology because this type of article - review - doesn't have typical methodology.  
  5. Are the conclusions consistent with the evidence and arguments
    presented and do they address the main question posed? In my opinion conclusions are consistet with the evidence and arguments presented and they address the main question posed.  
  6. Are the references appropriate? In my opinion references are appropriate.  
  7. Please include any additional comments on the tables and figures. The tables and figures are legible and cosistent with the description in the text.    

Author Response

(The authors gave the same response as above.)
